# Chromosome-associated RNA–protein complexes promote pairing of homologous chromosomes during meiosis in *Schizosaccharomyces pombe*

Da-Qiao Ding [1]*, Kasumi Okamasa[1], Yuki Katou[2], Eriko Oya [3,6], Jun-ichi Nakayama [3,4], Yuji Chikashige[1], Katsuhiko Shirahige[2], Tokuko Haraguchi [1,5] & Yasushi Hiraoka [1,5]*

Pairing of homologous chromosomes in meiosis is essential for sexual reproduction. We have previously demonstrated that the fission yeast *sme2* RNA, a meiosis-specific long noncoding RNA (lncRNA), accumulates at the *sme2* chromosomal loci and mediates their robust pairing in meiosis. However, the mechanisms underlying lncRNA-mediated homologous pairing have remained elusive. In this study, we identify conserved RNA-binding proteins that are required for robust pairing of homologous chromosomes. These proteins accumulate mainly at the *sme2* and two other chromosomal loci together with meiosis-specific lncRNAs transcribed from these loci. Remarkably, the chromosomal accumulation of these lncRNA–protein complexes is required for robust pairing. Moreover, the lncRNA–protein complexes exhibit phase separation properties, since 1,6-hexanediol treatment reversibly disassembled these complexes and disrupted the pairing of associated loci. We propose that lncRNA–protein complexes assembled at specific chromosomal loci mediate recognition and subsequent pairing of homologous chromosomes.

[1] Advanced ICT Research Institute Kobe, National Institute of Information and Communications Technology, Kobe 651-2492, Japan. [2] Institute for Quantitative Biosciences, The University of Tokyo, Tokyo 113-0032, Japan. [3] Graduate School of Natural Sciences, Nagoya City University, Nagoya 467-8501, Japan. [4] Division of Chromatin Regulation, National Institute for Basic Biology, Okazaki 444-8585, Japan. [5] Graduate School of Frontier Biosciences, Osaka University, Suita 565-0871, Japan. [6] Present address: Faculty of Science and Engineering, Chuo University, Tokyo 112-8551, Japan. *email: ding@nict.go.jp; hiraoka@fbs.osaka-u.ac.jp

Meiosis is an essential process for sexual reproduction in eukaryotes, that generates inheritable haploid gametes from a parental diploid cell. In this process, pairing of homologous chromosomes during the meiotic prophase and the resulting recombination-mediated physical links between homologous chromosomes are essential for the correct segregation of chromosomes during subsequent meiotic divisions[1,2]. Understanding the mechanisms involved in the pairing and recombination of homologous chromosomes is clinically important because chromosome mis-segregation during meiosis is a major cause of miscarriage and developmental abnormalities in humans[3].

During the pairing process, each homologous pair of chromosomes is aligned selectively. It is suggested that a bouquet arrangement of chromosomes, wherein chromosomes are bundled at telomeres to form a polarized configuration, contributes to the pairing of homologous chromosomes by spatially aligning them[4,5]. However, the mechanism by which homologs recognize and pair with their partner remains unclear. In different organisms, homologous chromosome pairing can be DNA double-strand break (DSB)-dependent or independent[2]. Even in organisms such as mouse, *Sordaria*, and budding yeast that mostly undergo DSB-dependent pairing, DSB-independent pairing can also be observed, suggesting the existence of mechanisms that perform homology matching and homolog pairing in the context of intact chromatin[2,6,7].

The fission yeast, *Schizosaccharomyces pombe*, exhibits a striking example of the bouquet arrangement. In this organism, the nucleus elongates (called a horsetail nucleus) and moves back and forth between the ends of a cell and the telomeres remain clustered at the leading edge of the moving nucleus[8,9]. The period of nuclar movements approximately corresponds to the meiotic prophase, but typical stages of the meiotic prophase are not charaterized; thus, this period is collectively called the horsetail stage. No synaptonemal complex forms in *S. pombe* and the pairing and recombination of homologous chromosomes occur during the horsetail stage. Homologous pairing observed in live meiotic cells demonstrated that telomere clustering and oscillatory chromosome movements spatially align homologous chromosomes in early stages of the meiotic prophase to promote their contact[10].

Furthermore, *S. pombe* also demonstrate recombination-independent pairing of homologous chromosomes, in which the robust pairing is mediated by noncoding RNA accumulated at the specific *sme2* gene locus[11,12]. The *sme2* RNA (also called meiRNA[13]) is a meiosis-specific, 1500-nt long noncoding RNA (lncRNA)[11,12]. It accumulates at its gene locus and plays an active role in recombination-independent pairing, leading to robust pairing[11]. However, the underlying mechanisms and their general impact remain to be elucidated. In this study, we identified a group of protein factors required for robust pairing and examined the mechanisms by which these protein factors together with lncRNA mediate the pairing of homologous chromosomes during meiosis. We demonstrate the roles of lncRNA–protein complexes assembled at specific chromosomal loci to tether homologous chromosomes.

## Results

**Identification of protein factors involved in robust pairing.** To elucidate the general mechanisms for *sme2* RNA-mediated homologous pairing, we first searched for proteins associated with *sme2* RNA by microscopic screening of the published GFP/YFP–protein fusion libraries[14–16]. In this screening, we searched for strains bearing nuclear dots during the meiotic prophase and identified 20 such strains. To search for nuclear dots localized at

the *sme2* locus, these strains were crossed with a strain carrying Mei2-mCherry, a well-known protein localized at the *sme2* locus[11,17]. Ten of the resultant strains displayed several nuclear dots, ranging from 1 to 6 (Fig. 1a; Supplementary Fig. 1a, b), one of which was co-localized with the Mei2 dot (yellow arrows in Fig. 1a). These proteins were designated as Smp1–Smp10 (*sme2* RNA-associated protein; Smp) (Table 1). All these Smp proteins are RNA-binding proteins commonly involved in RNA polyadenylation or transcription termination and are required for fundamental cellular functions in eukaryotes; their orthologs are conserved in a wide variety of eukaryotes including human (Table 1).

For functional analysis of these Smp proteins, six Smp proteins that are non-essential for growth (Rcd1, Nab3, Rmn1, Pab2, Rhn1, and Ctf1) were deleted; two of the four growth-essential Smp proteins (Rna15 and Pcf11) were downregulated only during meiosis, using auxin-inducible protein degradation (AID) system[18] (see the "Methods" section and Supplementary Fig. 2a, b); for *seb1*, a hypomorphic mutant *seb1-E38* was used (see the "Methods" section and Supplementary Fig. 2c–f). To examine meiotic pairing at the *sme2* locus in strains lacking Smp proteins, we measured distances between the homologous *sme2* loci detected by LacI-GFP signals during progression through the horsetail stage. The pairing frequencies were plotted for five substages (I–V) of the horsetail stage (~142 min in total[10]). These substages were divided equally, from karyogamy to the end of the horsetail movement in each cell. The *sme2* loci of distances ≤0.35 μm were counted as paired, following previous reports[10,11] (Fig. 1b). Percentile rank plots of the distance distribution are shown in Supplementary Fig. 3. The results showed that six Smp proteins (Rhn1, Pab2, Ctf1, Seb1, Pcf11, and Rna15) were required for robust pairing at the *sme2* locus (Fig. 1b, left and middle panels). These Smp proteins co-localized with Seb1 in the horsetail nucleus (Supplementary Fig. 1c). The remaining three Smp proteins did not affect the pairing frequency (Fig. 1b, right panel).

Since the *sme2* mutants defective in RNA dot formation are also defective in robust pairing[11], we examined whether Smp proteins affect the formation of *sme2* RNA dots. We counted the number of *sme2* RNA dots in the horsetail nuclei of Smp-mutant cells (Fig. 1c). In wild-type cells, the *sme2* RNA is predominantly confined to a single dot (see WT in Fig. 1c, right panel). In comparison, the absence of Smp proteins produced multiple dots of the *sme2* RNA; in particular, the absence of Rhn1 and Pab2 resulted in the striking dispersion of *sme2* RNA dots (Fig. 1c, right). These results indicate that Smp proteins mediate the chromosomal accumulation of *sme2* RNA dots, which correlates with robust pairing.

**Identification of chromosomal loci for the accumulation of protein factors.** Interestingly, 9 of the 10 Smp proteins (Smp2–Smp10) showed two or more additional nuclear dots besides the *sme2* locus (Fig. 1a). As these additional dots may represent new robust-pairing sites, we attempted to determine their chromosomal positions. To identify sites corresponding to the nuclear dots, we followed a cytological approach using a genome-wide *lacO* insertion library. In this approach, we constructed a *lacO* insertion library along the whole genome, marking 143 loci with an average interval of ~90 kbp (Fig. 2a, Supplementary Table 1). Using this library, we examined the co-localization of LacI-GFP signals with Seb1-mCherry. We found two locations where the LacI-GFP signal was co-localized with the Seb1-mCherry dot, namely, the A55 site on chromosome I (Fig. 2b) and the C24 site on chromosome III (Fig. 2c). We further constructed a strain containing the *lacO* array at *sme2*,

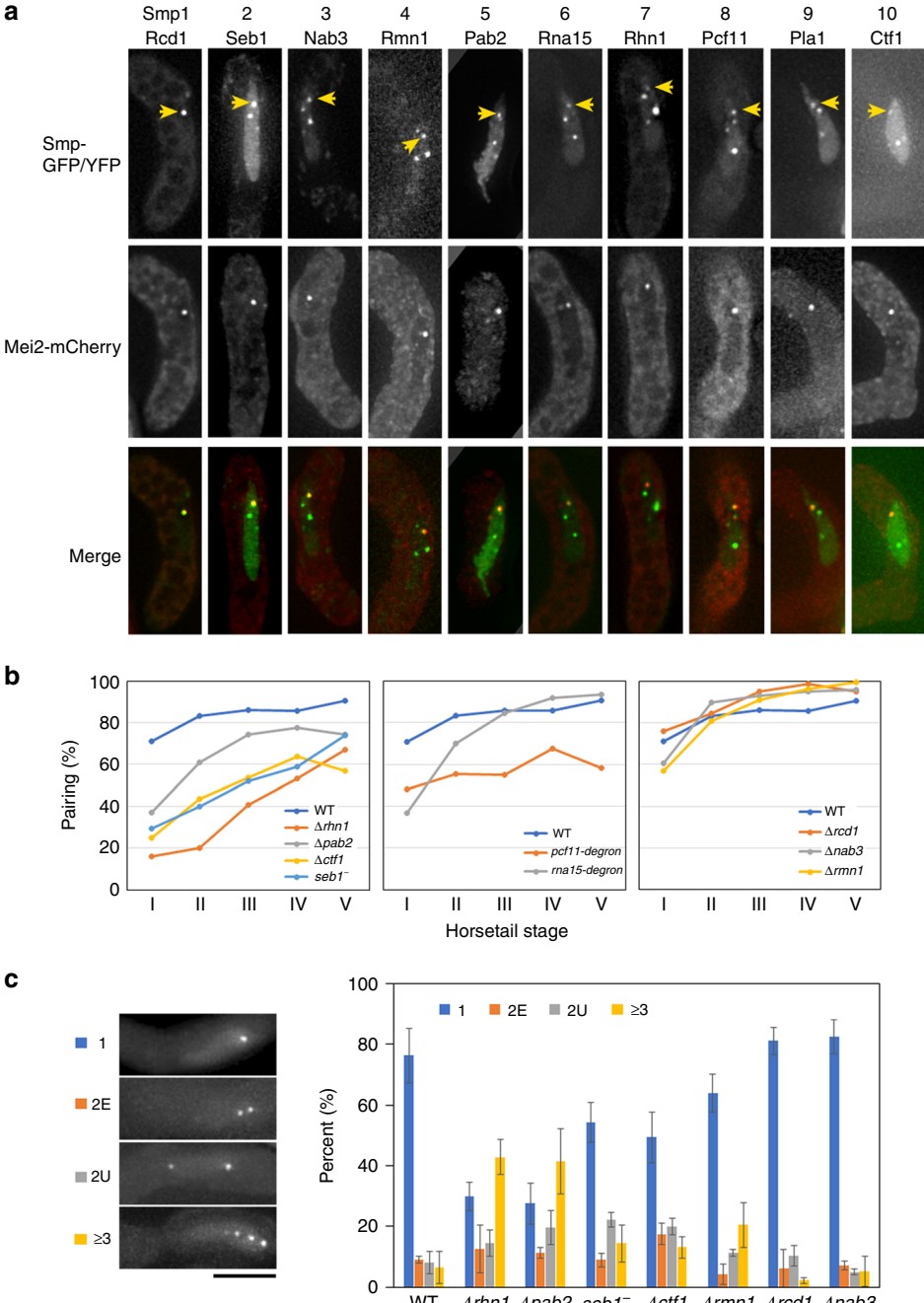

**Fig. 1 Smp proteins are required for robust pairing at the *sme2* locus. a** Localization of Smp1–Smp10 proteins fused with GFP or YFP in the horsetail nucleus. Arrows indicate co-localization with Mei2-mCherry. Meiosis was induced in the homothallic strains expressing YFP-fused or GFP-fused Smp protein and Mei2-mCherry. Three-dimensional fluorescence images of YFP/GFP and mCherry were acquired in live zygotes. Yellow arrows indicate the position of Smp protein co-localized with Mei2 at the *sme2* locus. Scale bar, 5 μm. **b** Pairing frequency (%) of the *sme2* locus in wild type (WT) and various defective mutants of Smp proteins: Δ*rhn1*, Δ*pab2*, Δ*ctf1*, *seb1-E38*, *pcf11-degron*, *rna15-degron*, Δ*rcd1*, Δ*nab3*, and Δ*rmn1*. The number of cells examined is provided in a Source Data file 1b. Characterization of the *pcf11-degron*, *rna15-degron*, and *seb1-E38* mutants, is shown in Supplementary Fig. 2. Cells of all the strains were induced to meiosis at 26 °C. The *sme2* loci at distance ≤0.35 μm were counted as paired, and pairing frequencies were plotted for five substages (I–V) of the horsetail stage. Percentile rank plots of the distance between homologous loci are shown in Supplementary Fig. 3. Source data for Fig. 1b and Supplementary Fig. 3 are provided as a Source Data file 1b. **c** Frequency (%) of the number of *sme2* RNA dots in WT and various Smp-defective mutant cells at the horsetail stage. The RNA dots were classified into four classes (1 for a single dot, 2E for 2 RNA dots with equal size, 2U for 2 RNA dots with unequal size, and >3 for more than 2 dots). Representative images for one to three or more dots are shown on the left. Two dots with equal size (2E) or unequal size (2U) were counted separately. Class 2E likely represents *sme2* RNA accumulated at each of the homologous *sme2* loci. More than 60 horsetail nuclei in an asynchronous population were observed for each strain, in three independent experiments; the precise number of cells examined is provided in a Source Data file 1c. Error bars indicate standard deviation. Scale bar, 5 μm. Source data for Fig. 1c are provided as a Source Data file 1c.

**Table 1 List of Smp proteins.**

| Smp protein | Gene product | Human orthologs | Localization in meiotic prophase | Gene deletion viability | Robust pairing requirement |
|---|---|---|---|---|---|
| Smp1: Rcd1 (SPAC29B12.06c) | CCR4–NOT complex RNA-binding protein subunit | CNOT9 | Mei2-dot | Viable | No |
| Smp2: Seb1 (SPAC222.09) | RNA-binding and 3′-end processing protein | SCAF8 | Mei2-dot, Nuclear dots | Inviable | Yes |
| Smp3: Nab3 (SPAC3H8.09c) | Nrd1 complex poly(A) binding protein | RALY, RALYL, HNRNPC, HNRNPCL1-4 | Mei2-dot, Nuclear dots | Viable | No |
| Smp4: Rmn1 (SPBC902.04) | RNA-binding protein | RBM26, RBM27 | Mei2-dot, Nuclear dots | Viable | No |
| Smp5: Pab2 (SPBC16E9.12c) | poly(A)-binding protein | PABPN1L, PABPN1 | Mei2-dot, Nuclear dots | Viable | Yes |
| Smp6: Rna15 (SPAC644.16) | RNA-binding protein | CSTF2T, CSTF2 | Mei2-dot, Nuclear dots | Inviable | Yes |
| Smp7: Rhn1/Iss4 (SPBC337.03) | RNA polymerase II transcription termination factor homolog | RPRD1B, RPRD1A | Mei2-dot, Nuclear dots | Viable | Yes |
| Smp8: Pcf11 (SPAC4G9.04c) | mRNA cleavage and polyadenylation specificity factor | PCF11 | Mei2-dot, Nuclear dots | Inviable | Yes |
| Smp9: Pla1 (SPBC646.04) | poly(A) polymerase | PAPOLA, PAPOLG, PAPOLG | Mei2-dot, Nuclear dots | Inviable | N.A. |
| Smp10: Ctf1 (SPBC3B9.11c) | mRNA cleavage and polyadenylation specificity factor complex subunit | CSTF2T, CSTF2 | Mei2-dot, Nuclear dots | Viable | Yes |

A55, and C24, and confirmed that Seb1-GFP co-localized with each of the three loci in the same cell (Fig. 2d). Thus, we identified three sites of Smp protein chromosomal accumulation, one on each of the three chromosomes (green arrows in Fig. 2a).

**Identification of lncRNA required for the accumulation of protein factors.** The precise binding sites of Smp foci on the genome were determined by ChIP-seq (chromatin immunoprecipitation followed by DNA sequencing) analysis of GFP-tagged Rhn1, Pab2, and Seb1, in vegetative cells and in meiotic prophase cells. Their binding sites on the genome are shown in Supplementary Data 1. ChIP-seq profiles of Rhn1, Pab2, and Seb1 revealed a binding peak at the *sme2* locus as expected, along with the following two binding peaks common to these three Smp proteins: the A55 site was narrowed down to approximately a 10-kb region at the *mei2* locus, and C24 was narrowed down to the 3′-end of the *ski3* locus (Fig. 2e).

To identify the gene responsible for the formation of Smp foci, the nearby genes (*mei2* and *omt3*) were deleted. The Smp dot at the A55 locus remained when *mei2* was deleted (Fig. 2f, left), but disappeared when *omt3* (SPNCRNA.130) was deleted (Fig. 2f, right). Quantification showed that the number of Seb1 dots decreased in the *omt3* deletion compared to the *mei2* deletion (Supplementary Fig. 4a), indicating that *omt3* is responsible for the formation of Smp foci at A55. *omt3* encodes a meiosis-specific lncRNA (1001 nt) with an unknown function[19]. Similarly, the Smp dot at the C24 locus remained when *ski3* was deleted (Fig. 2g, left), but disappeared when a longer fragment extending to the 3′-UTR of *ski3* was deleted (Fig. 2g, right). Quantification showed that the number of Seb1 dots in the longer deletion were decreased compared to those with the *ski3* deletion (Supplementary Fig. 4b). This longer fragment contains the *SPNCRNA.584* gene, encoding a meiosis-specific lncRNA (annotated as an antisense RNA of *ski3*; 2229 nt)[20], indicating that this lncRNA (hereafter, lncRNA584) is responsible for the formation of an Smp dot at this locus. The number of Seb1 dots decreased to 1 dot or less in all three lncRNA deletions (Supplementary Fig. 4c), indicating that Seb1 dot formation mainly depends on these lncRNA species. Taken together, we identified three meiosis-specific lncRNAs (*sme2* RNA, *omt3* RNA, and lncRNA584) that are required for the formation of Smp foci. No obvious sequence similarity was detected among these three lncRNA species.

**lncRNA species accumulate at their gene loci on the chromosome.** We then determined the localization of the newly found lncRNA species using single-molecule fluorescence in situ hybridization (smFISH). As previously observed using fluorescently tagged *sme2* RNA in living cells[11], the *sme2* RNA formed a single RNA dot at the *sme2* locus (Fig. 3a). Similarly, the *omt3* RNA and lncRNA584 also formed a single RNA dot at each of their respective gene loci (Fig. 3b, c). The following three major patterns of lncRNA localization were observed with respect to the gene locus: two RNA dots associated with each LacI-GFP signal (labeled as s2 for two separated dots, Fig. 3a–c), a single RNA dot localized between two juxtaposed LacI-GFP signals (p2 for paired RNA dot with two LacI dots, Fig. 3a–c), and a single RNA dot on a single LacI-GFP signal (p1 for one paired dot, Fig. 3a–c). In the smFISH experiments, the distance between two homologous loci connected by a single RNA dot was 0.53 μm on average (sd = 0.096, 32 cells). Because the cells shrank to 79% by fixation compared to live cell, this distance corresponds to 0.67 μm in live cells. Therefore, based on the pairing distance of RNA dots, the LacI-GFP signals with a distance of ≤0.67 μm (p1 and p2 states) were hereafter regarded as paired. These observations suggest that the foci of RNA–protein complex accumulated on each

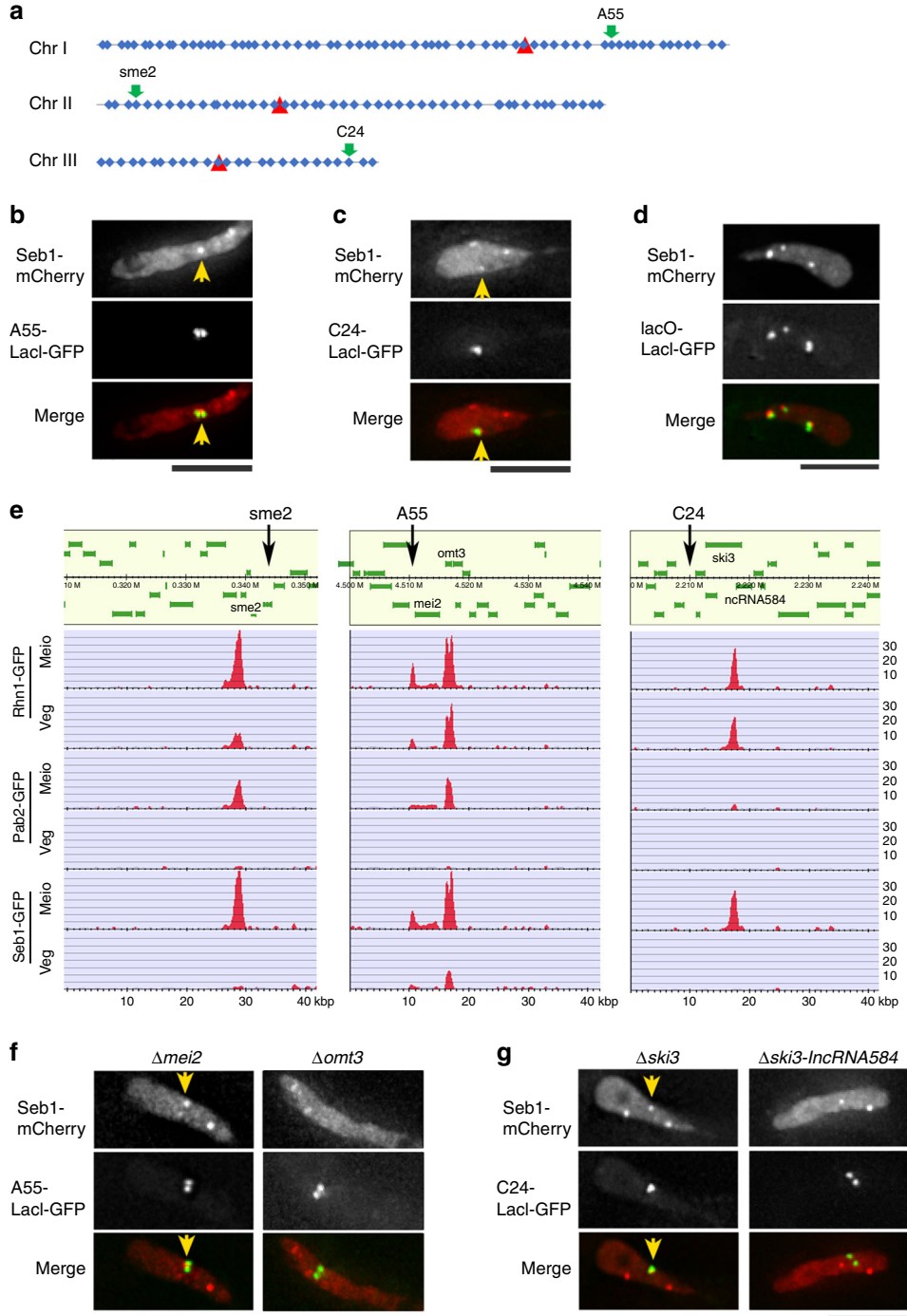

**Fig. 2 Identification of Smp dots and their role in homologous pairing. a** A map of a *lacO* insertion library of the *S. pombe* genome, composed of the following three chromosomes: average intervals of 82 kbp on chromosome I (Chr I), 90 kpb on chromosome II (Chr II), and 95 kbp on chromosome III (Chr III). Blue diamonds show the loci for *lacO* insertion. Red triangles indicate the position of the centromere. The complete list of *lacO*-insertion loci is provided in Supplementary Table 1. **b**, **c** Localization of A55-*lacO*/LacI-GFP **b** or C24-*lacO*/LacI-GFP **c** with one of the Seb1-mCherry dots. Arrows indicate the co-localized position. **d** Co-localization of Seb1-mCherry dots with *lacO*/LacI-GFP at the A55, C24, and *sme2* loci. **e** ChIP-seq data of Rhn1, Pab2, and Seb1 in meiotic (Meio) and vegetative (Veg) cells. Arrows indicate the *lacO*-insertion sites at *sme2* (left), A55 (middle), and C24 (right). The map of the *S. pombe* genome was obtained from the PomBase public database (https://www.pombase.org). The complete data from the genome-wide ChIP-seq are shown in Supplementary Data 1. **f** Localization of A55-*lacO*/LacI-GFP and Seb1-mCherry in *mei2* deletion (left) and *omt3* deletion (right) mutants. Arrows indicate the co-localized position. **g** Localization of C24-*lacO*/LacI-GFP and Seb1-mCherry in Δ*ski3* (left) and Δ*ski3-lncRNA584* (right) mutants. Arrows indicate the co-localized position. Scale bar, 5 μm **b**, **c**, **d**, **f**, **g**.

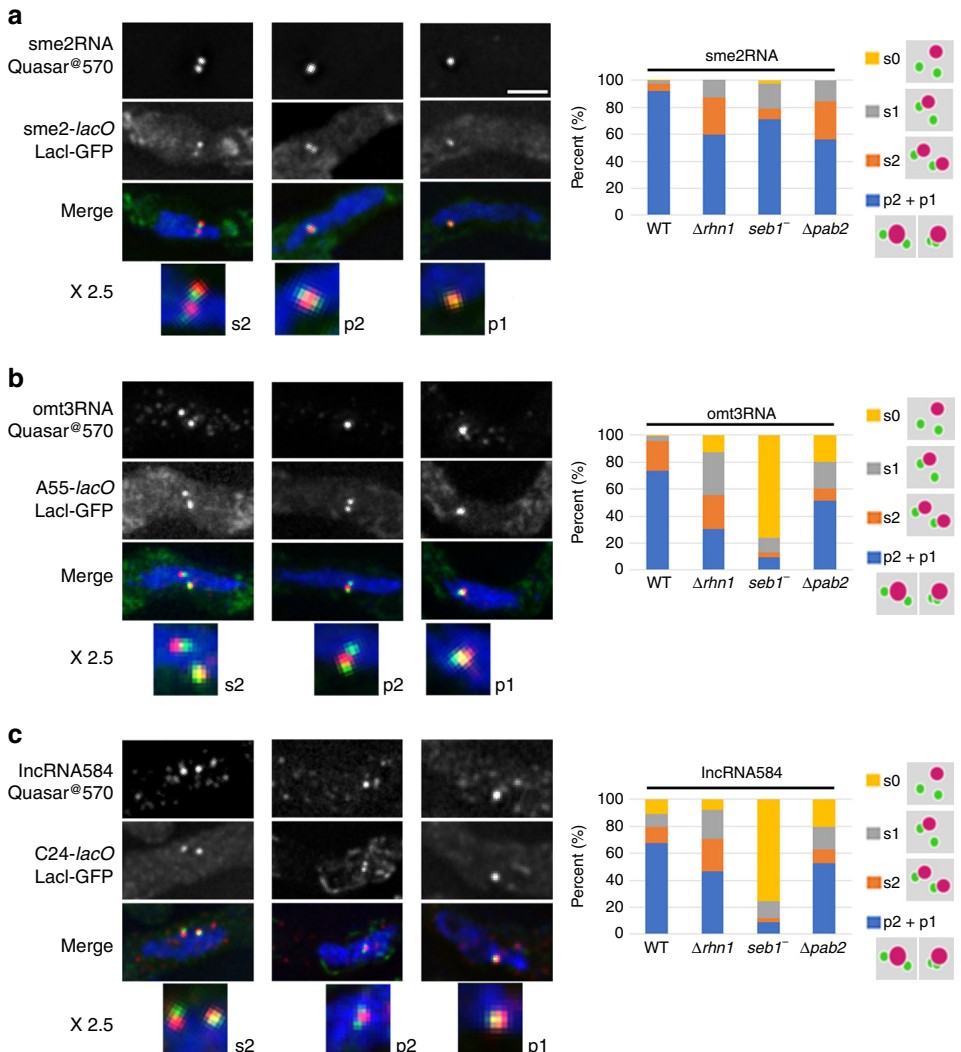

**Fig. 3 Localization of lncRNA on homologous chromosome loci.** (Left) smFISH analysis for *sme2* RNA (**a**), *omt3* RNA (**b**), and lncRNA584 (**c**) using sequence-specific probes labeled with Quasar®570. Chromosomal loci were visualized using *lacO*/LacI-GFP. DNA was visualized using 4',6-diamidino-2-phenylindole. In the merged images, smFISH is shown in red, *lacO*/LacI-GFP in green, and DNA in blue. Enlarged images (×2.5) are shown at the bottom. The abbreviations are as follows: s2: two separate RNA dots, each associated with a LacI-GFP dot, p2: a single RNA dot between two juxtaposed LacI-GFP dots, and p1: a single RNA dot with a paired LacI-GFP dot. Scale bar, 2 μm. (Right) Signal patterns of smFISH counted (*n* > 100 for each sample) in WT, *rhn1* deletion mutant (*Δrhn1*), *seb1-E38* mutant (*seb1⁻*), and *pab2* deletion (*Δpab2*) mutant. p2 and p1 (blue) and s2 (orange) are defined above. s1 (gray): two separate LacI-GFP dots, only one of which is associated with an RNA dot. s0 (yellow): two separate LacI-GFP dots, none of which is associated with an RNA dot. Source data for Fig. 3 are provided as a Source Data file 3.

homologous locus fuse together to form a single focus, connecting the juxtaposed homologous loci. In mutants lacking the Smp proteins, paired RNA dots (p1 and p2) were decreased dramatically (Fig. 3a–c, right panels), and the number of RNA dots that were not associated with their genetic loci, which were rarely observed in wild type, increased (s1 and s0 in Fig. 3a–c, right panels). Especially, *omt3* RNA and lncRNA584 depended on Seb1 more strongly than the *sme2* RNA for their chromosomal accumulation (Fig. 3a–c, right panels). These observations suggest the importance of Smp proteins in the chromosomal accumulation of lncRNA, which in turn is important for lncRNA-mediated homologous pairing.

**Chromosomal lncRNA–protein complexes mediate pairing of homologous loci.** We measured the pairing frequency at the newly identified loci (A55 and C24) in the horsetail substages I–V to compare with that at the *sme2* locus (Fig. 4). Pairing frequency was determined with a distance of ≤0.67 μm as paired based on

the pairing distance of RNA dots. Percentile rank plots of the distance distribution are shown in Supplementary Fig. 5. We assessed the contributions of lncRNA to pairing with or without recombination. The A55 and C24 loci showed a similar pairing frequency with (WT) or without recombination (*rec12⁻*) during the early horsetail stages (horsetail stages I–III), whereas the pairing frequency was decreased without recombination in horsetail stages IV and V (Fig. 4a and Supplementary Fig. 5), suggesting the contributions of recombination to pairing at later stages. When *omt3* was deleted, A55 pairing decreased at the early horsetail stages (*Δomt3* in Fig. 4a and Supplementary Fig. 5), and became recombination-dependent as pairing was strikingly inhibited in the *rec12* and *omt3* double mutant (*rec12⁻ Δomt3* in Fig. 4a and Supplementary Fig. 5). Depletion or mutation of Rhn1 and Seb1 decreased the pairing frequency at the A55 locus (Fig. 4b). Thus, omt3RNA promotes recombination-independent pairing at the A55 locus in an Smp-dependent manner. In

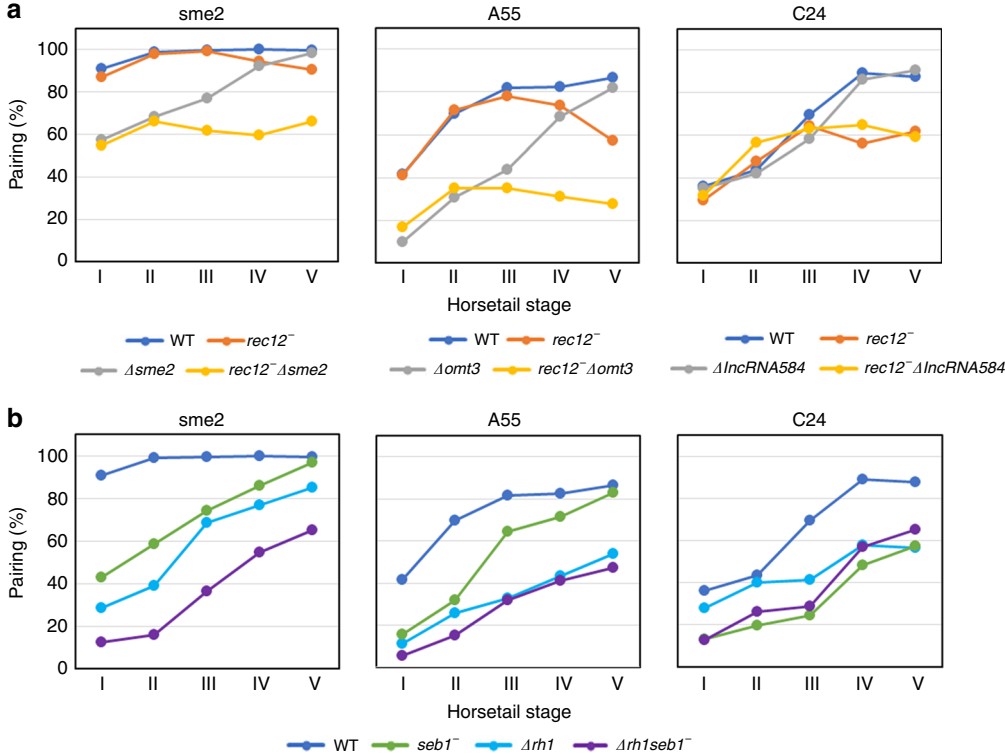

**Fig. 4 Homologous chromosome pairing at lncRNA accumulation sites. a, b** Pairing frequencies were plotted for five substages (I–V). Pairing frequency (%) at the *sme2*, A55, and C24 loci in WT (blue), *rec12⁻* (orange), Δ*sme2*, Δ*omt3,* and Δ*lncRNA584* (gray), Δ*sme2 rec12⁻*, Δ*omt3 rec12⁻*, and Δ*ncRNA.584 rec12⁻* (yellow) Δ*rhn1* (light blue), *seb1⁻* (green), and Δ*rhn1 seb1⁻* double mutants (purple) was measured in 18–53 cells for each strain; the precise number of cells examined for each strain is provided in Source Data file 4. Percentile rank plots of the distance between homologous loci are shown in Supplementary Fig. 5. Source data for Fig. 4 and Supplementary Fig. 5 and are provided as a Source Data file 4.

contrast, lncRNA584 had limited contributions to pairing at C24 in both recombination positive or negative background (Δ*lncRNA584* in Fig. 4a and Supplementary Fig. 5). The depletion or mutation of Rhn1 and Seb1 decreased the pairing frequency at the C24 locus (Fig. 4b), indicating that C24 pairing requires Smp proteins. Therefore, it is likely that Smp-dependent pairing at nearby regions contributes to pairing at the C24 locus. The C24 locus is close to the rDNA repeat cluster located at both ends of Chromosome III, which form the nucleolus. Pairing of chromosome III is likely dominated by the gigantic RNA body of the nucleolus, underrating the contributions of lncRNA584 to the pairing of homologous chromosomes. Dependency of pairing on Seb1 and Rhn1 was different among the *sme2*, A55 and C24 loci: Rhn1 dominated pairing at the A55 and Seb1 dominated pairing at the C24 locus, whereas Seb1 and Rhn1 synergistically acted on the *sme2* locus pairing. Thus, the contributions of Smp proteins and lncRNA to pairing can differ from one locus to another. Depletion or mutation of Seb1, Rhn1, and Pab2 showed no serious defects in chromosome segregation, spore formation or spore viability except for low spore viability in Δ*rhn1* (Supplementary Fig. 2g).

**Liquid droplet properties of Smp proteins.** In smFISH staining of these three lncRNA species, they always appear as a round shape with different sizes (p2 and p1 in Fig. 3a–c). Furthermore, no two RNA dots touching each other were observed, implying an immediate fusion to one dot. As this appearance and behavior of RNA dots seemed analogous to liquid droplets and their fusion, we hypothesized that phase separation might be involved in the recognition of homologous loci. To test this hypothesis, we

continuously observed the foci of Smp proteins and lncRNA in living cells; during live observation, cells were briefly (3 min) treated with 1,6-hexanediol to resolve the compartments of phase separation[21–23]. The Smp protein foci disappeared upon the addition of 1,6-hexanediol and reappeared after its removal (Fig. 5a; Supplementary Fig. 6a, b; Supplementary Movies 1–3). Accordingly, a single focus of the *sme2* RNA scattered to multiple foci upon the addition of 1,6-hexanediol, and the single focus was restored after its removal (Fig. 5b, Supplementary Movie 4). Furthermore, the paired loci of *lacO* insertion were separated upon the addition of 1,6-hexanediol and re-paired after its removal (Fig. 5c; Supplementary Fig. 6c; Supplementary Movies 5, 6). These cells proceeded with normal meiotic nuclear divisions and sporulation after the removal of 1,6-hexanediol. The pairing frequencies at the *sme2*, A55, and C24 loci upon 1,6-hexanediol treatment were plotted against time (Fig. 5d). The pairing frequency at these loci was decreased upon the addition of 1,6-hexanediol and gradually recovered after the removal of 1,6-hexanediol. A decrease in the pairing frequency was much more striking in recombination-negative cells than in the wild type cells (Fig. 5d, A55 and C24). As a control, Rec8 remained on chromatin with 1,6-hexanediol treatment (Supplementary Fig. 6d), confirming that the meiotic chromosome axis was not destroyed. These results suggest a significant contribution of the phase separation of Smp proteins to recombination-independent pairing as depicted in Fig. 6c.

**lncRNA determines specificity for pairing of homologous loci.** To elucidate factors that confer specificity for homologous recognition, we tested pairing of the A55 locus in a heterozygous

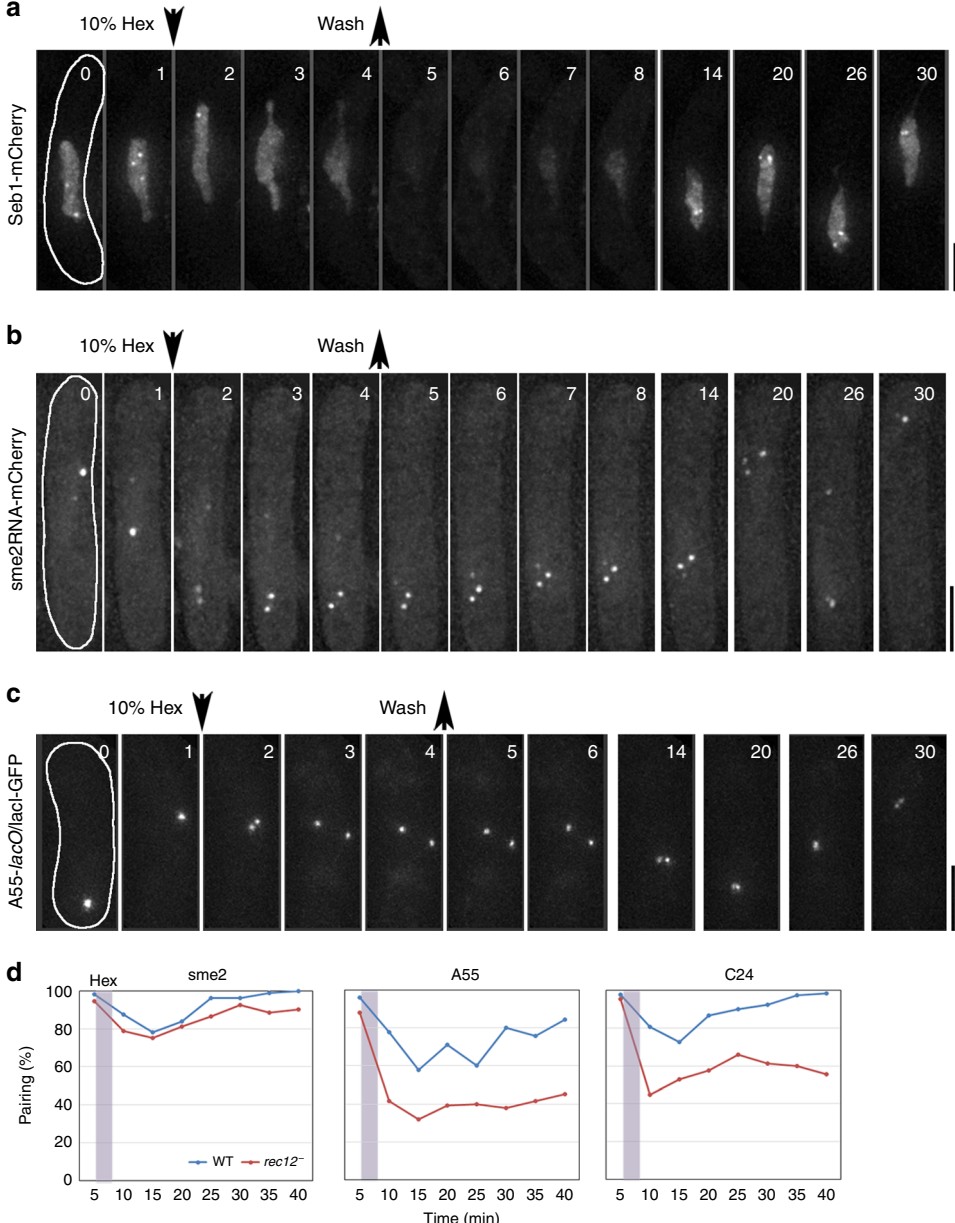

**Fig. 5 Liquid-phase separation of Smp proteins and its impact on homologous pairing. a–c** Selected projected time-lapse images of live cells at the horsetail stage for Seb1-mCherry (**a**), *sme2* RNA-U1Atag/U1Ap-mCherry (**b**), and A55-*lacO*/LacI-GFP (**c**). The *sme2* RNA-U1Atag was visualized by U1Ap-mCherry[11]. Three-dimensional images were captured every minute. Microfluidic yeast plates (CellASIC) were used for cell culture on the microscope stage. 1,6-Hexanediol (10%) was added or removed, as indicated by the arrows. The solution in the cell culture chamber was changed at <1 min. Scale bar, 5 μm. **d** Number (%) of cells with paired chromosomes at *sme 2* (left), A55 (middle), and C24 (right) loci. 1,6-Hexanediol was added to WT and *rec12* deleted (*rec12⁻*) cells at 5 min and removed at 8 min (treatment period is shaded in the graphs). Three-dimensional live cell images were captured every minute, and the percentage of paired cells observed at 5-min intervals is shown. Source data for Fig. 5 are provided as a Source Data file 5.

strain expressing the *sme2* RNA and the *omt3* RNA from the A55 locus on each of the homologous chromosomes. Pairing frequency at the A55 locus was decreased in the heterozygous cross (*omt3* × *sme2* in Fig. 6a) to a level similar to that in the homozygous deletion of *omt3* (Δ*omt3* in Fig. 6a) in contrast to the WT and a homozygous strain expressing the *sme2* RNA from the *omt3* locus on both homologous chromosomes (WT and Δ*omt3*:: *sme2* in Fig. 6a). When the *sme2* RNA was expressed from the A55 locus in the homozygous strain, it formed RNA dots at the A55 locus on each of the homologous chromosomes, and these RNA dots were fused effectively similar to those expressed for the

*sme2* locus (Fig. 6b, middle row of panel). However, when the *sme2* RNA and *omt3* RNA were expressed from the A55 locus in the homozygous strain, the *sme2* RNA dots and the *omt3* RNA dots were formed on each of the homologous chromosomes, but were barely fused to a single dot, even when they were located adjacent to each other (Fig. 6b, third row of panel). The numbers of RNA dots were counted in the nuclei showing the LacI-GFP signals ≤0.67 μm in three-dimensional data sets. Homozygous strains expressing the *omt3* RNA or the *sme2* RNA displayed a single RNA dot in 93.7% or 98.2% of the nuclei, respectively, whereas heterozygous strain expressing the *sme2* RNA and the *omt3* RNA displayed two

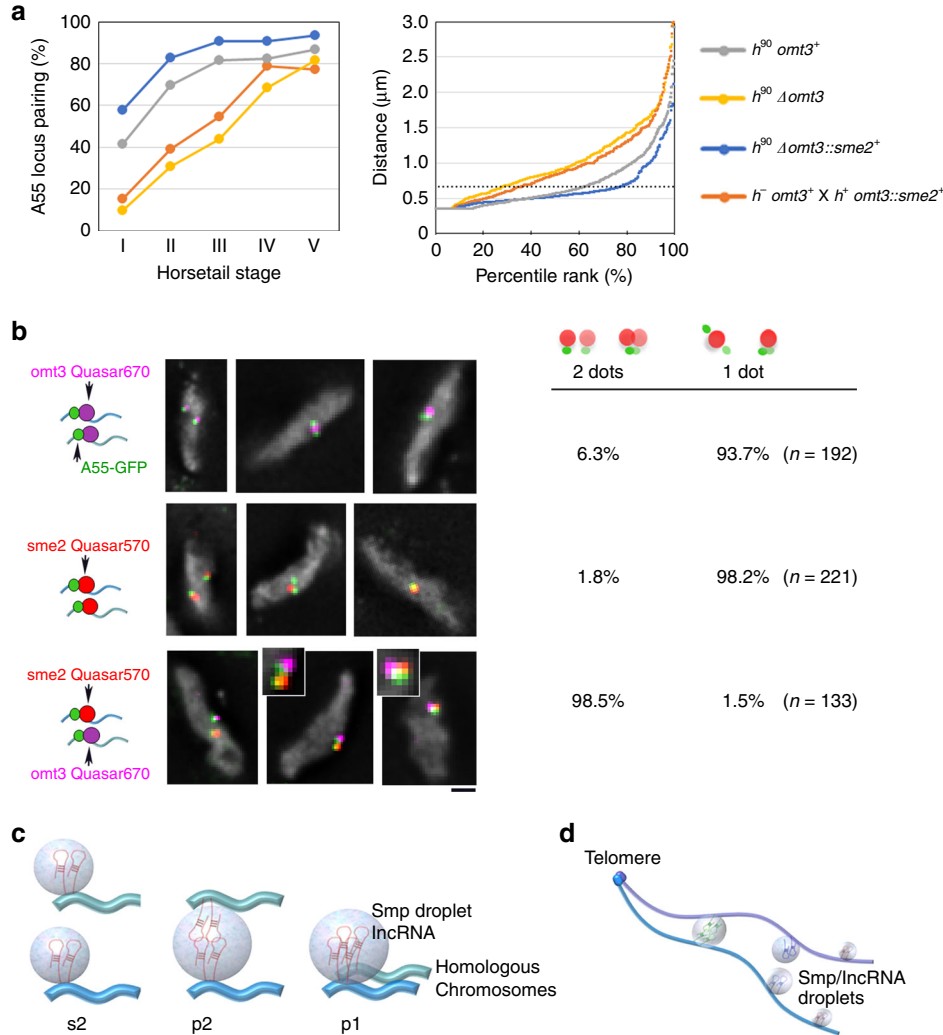

**Fig. 6 Specificity for the recognition of homologous loci. a** Pairing frequency at the A55 locus in $h^{90}$ homozygous zygote cells (left panel): wild type (gray), $\Delta omt3$ (yellow), replacement of $omt3$ with $sme2$ ($\Delta omt3::sme2^+$, blue); and in heterozygous zygote cells from a wild type $h^-omt3^+$ crossed with a $h^+$ $\Delta omt3::sme2^+$ strain (orange). Right panel shows percentile rank plots of the distance between the A55 loci in the first three substages. The dotted line at 0.67 μm on the Y-axis indicates the threshold of pairing. The authentic $sme2$ gene was deleted in $h^{90}$ $\Delta omt3::sme2^+$ cells, and in the heterozygous crossed zygote cells. Source data for Fig. 6 are provided as a Source Data file 6. **b** smFISH analysis for $omt3$ RNA (probes labeled with Quasar®670, purple) and $sme2$ RNA (probes labeled with Quasar®570, red) for the strain combinations shown in **a**. Chromosomal loci were visualized using $lacO$/LacI-GFP (green). DNA was visualized using DAPI. Typical smFISH images in the horsetail nuclei are shown on the left; the bar represents 1 μm. On the left column, a summary of two categories of staining is shown: two GFP dots each with an RNA dot, separated from each other by <0.67 μm; or a paired GFP dot with one RNA dot. Number of horsetail nuclei examined are shown in the parentheses. **c** Schematic representation of the pairing of homologous loci in this study. Blue lines represent chromosomes and light blue spheres represent RNA bodies containing Smp proteins and lncRNAs. The s2, p2, and p1 states are defined as in Fig. 3. **d** The model of chromosome pairing. Blue and purple lines represent a set of homologous chromosomes held together at the telomeres. Spheres represent liquid droplets containing Smp proteins and lncRNAs. Arrays of phase-separated Smp/lncRNA droplets align due to telomere clustering. The homologous droplets fuse together upon contact, promoting the pairing of homologous chromosomes.

separate dots of RNA in 98.5% of nuclei (Fig. 6b). These data suggest that RNA species in the Smp droplet play a major role in determining the identity of the RNA–Smp droplet.

## Discussion

We previously identified $sme2$ lncRNA, which is accumulated at the $sme2$ gene locus and is required for pairing of homologous chromosomal loci of $sme2$[11]. The mechanisms by which lncRNA is retained at the chromosomal locus of its transcription site are not known. In this study, we identified protein factors (Smp proteins) that are required for the accumulation of $sme2$ RNA and other lncRNA species. Three of the Smp proteins (Seb1, Rhn1, and Pcf11) contain a so-called CID domain (CID, CTD-

interacting domain; CTD, C-terminal domain of RNA polymerase II), suggesting the involvement of RNA transcription and termination, a fundamental machinery, in the chromosomal accumulation of lncRNA. Recent studies have revealed that Seb1 is required for the termination of both coding and noncoding transcripts through interaction with the RNA polymerase II (Pol II) C-terminal domain and nascent RNA[24,25]; it also causes pausing of Pol II, which facilitates heterochromatin assembly of the centromere[26]. Therefore, Smp proteins may retain lncRNA at their transcription loci through coupling with Pol II transcription termination. This is consistent with our previous finding that deleting the polyadenylation sites of $sme2$ RNA eliminated its chromosome accumulation[12].

Notably, the identified lncRNAs are all meiosis-specific transcripts. Some of the meiosis-specific RNAs are degraded during mitosis through a Mmi1-dependent mechanism, and Mmi1 binds to the determinant of selective removal (DSR) sequences contained in such RNAs, including sme2[27]. It is reported that 14 repeats of DSR can substitute the function of sme2 RNA to form Mmi1 dots[28]. The lncRNA species that we identified, also contain the DSR sequences. Therefore, it is possible that the DSR sequences are responsible for robust pairing. However, our experiments showed that 14 DSR repeats inserted at the A37 locus on chromosome I did not affect the pairing frequency of this locus nor the accumulation of Seb1 (Supplementary Fig. 7), making this possibility unlikely. It is also unlikely that the exosome is involved in lncRNA-mediated homologous pairing because omt3 RNA is not listed as targets of the exosome[29].

Our findings demonstrate that Smp proteins accumulate at three chromosome loci, one on each chromosome, and promote the pairing of homologous chromosomes. These major Smp-binding sites seem to serve as a pairing center as a striking example of pairing center has been observed in the nematode C. elegans, where loss of the paring center eliminates the pairing of the entire chromosomes[30]. However, loss of Smp-binding sites does not the eliminate pairing of entire chromosomes. Instead the previous study demonstrated that the effect of sme2 RNA on robust pairing is limited to a 200-kb region around the sme2 locus[11], suggesting that RNA-mediated robust pairing is local. Thus, we propose that a combination of local pairing sites along the chromosome contribute to the long-range recognition of homologous chromosomes for pairing. Multiple binding sites along the chromosome, as revealed by ChIP-seq analysis, serve as an array of pairing sites, analogous to a bar-code model proposing an array of transcription factories that juxtapose a pair of homologous chromosomes[31,32].

In this study, we demonstrated that the Smp protein factors mediate phase separation at lncRNA transcription sites. The Smp proteins contain intrinsically disordered regions necessary for phase separation. Phase separation has been observed in many biological phenomena[21,33,34]; for example, heterochromatin protein 1 plays a key role in heterochromatin formation through its phase separation[35,36] and phase separation of lncRNA drives paraspeckle and P granule formation[23,37]. Our finding that two lncRNA–Smp droplets can fuse only when they are bearing the same RNA species (Fig. 6b) may be related with the reported finding that the RNA sequence influences the physical properties of phase-separated protein–RNA droplets[38]. In conclusion, the present study shows that Smp proteins play a key role in phase separation and that lncRNAs play a role in determining the specificity of chromosomal loci for fusion.

Based on the present and previous studies, we propose a sequential process of homologous chromosome pairing. Chromosomes are bundled at telomeres and aligned in the oriented configuration (Fig. 6d)[4,5,39,40]. Smp-mediated recognition of homologous chromosomes occurs without recombination at the early stages. At these stages, nuclear movements agitate chromosomes to promote the pairing of homologous chromosomes, and simultaneously eliminate the undesired pairing of non-homologous chromosomes[10,40–42]. Finally, homologous chromosomes are stably connected by homologous recombination at later stages. Our findings provide insights into the mechanisms underlying recombination-independent recognition of homologous chromosomes, generally observed during sexual reproduction in eukaryotes. Telomere clustering in the meiotic prophase has been reported in a wide variety of eukaryotes[4,5,39,40]. Smp proteins are well-conserved RNA-processing proteins among eukaryotes, including humans (Table 1). Phase separation is a generic physicochemical property. Therefore, these mechanisms may be conserved in various eukaryotes.

## Methods

**Strains and culture media.** S. pombe strains used in this study are listed in Supplementary Table 2. S. pombe standard culture media YES, ME, and EMM2-N were used for routine culture, meiosis induction, and live observation, respectively[43]. Strains for live cell observation were constructed as follows: GFP and mCherry fusions were constructed using a PCR-based gene-targeting method[44], where the open-reading frame of GFP or mCherry was integrated at the C-terminal end of the endogenous gene locus in the genome. Strains for functional analysis constructed as follows. Six genes of Smp proteins that are non-essential for growth (Rhn1, Pab2, Ctf1, Rmn1, Rcd1, and Nab3) were deleted. Two of the four growth-essential Smp proteins (Pcf11 and Rna15) were downregulated only during meiosis, using the AID system[18]. AID strains were constructed as described previously[45]: briefly, IAA17-mCherry was integrated into the 3′-end of the native pcf11 or rna15 gene, and skp1-AtTIR1–2NLS-9myc was integrated into the lys1 gene locus. Auxin (0.5 mM) was added to the EMM2-N medium and during live cell observation. Protein depletion was confirmed by the disappearance of Smp-IAA17-mCherry fluorescence. For seb1, a hypomorphic mutant seb1-E38 was used. Primers used for gene disruption are listed in Supplementary Table 3.

**Live cell analysis of homologous chromosome pairing.** Microscopic observations were carried out on the DeltaVision Elite microscope system (GE Healthcare, Buckinghamshire, UK) using an Olympus oil-immersion objective lens PlanApoN60xOSC (NA = 1.4). Specific chromosome loci were visualized using the lac operator/lac repressor-GFP (lacO/LacI-GFP) recognition system[46]. To induce meiosis, vegetatively growing cells were transferred to ME plates for 8–12 h at 20–26 °C, depending on the strain background. Cells in the meiotic prophase were then suspended in EMM2-N medium supplemented with the appropriate amino acids for live observations. Live cell observation was carried out at 26 °C and data were collected to measure homologous chromosome pairing as described previously[10]: briefly, in measuring homologous chromosome pairing, we observed the same living cells continuously, and images were recorded every 5 min for 2–3 h during the horsetail stage. We divided the horsetail stage equally to five substages. The number of cells observed was 20–40 for each strain. Distances were measured in a data set of time-lapse images composed of typically five timepoints in 20–40 cells, making about 100–200 measurements in each substage for each strain. The precise number of measurements is provided in Source Data files 1b and 4. The number of measurements was normalized as a percentage, and the distance distribution was plotted as a percentile rank.

For counting the pairing frequency in Fig. 1b, the LacI-GFP signals with distance ≤0.35 μm (p1 state defined in Fig. 3) were regarded as paired, following previous reports[10,11]. For counting the pairing frequency in Figs. 2d and 4d, LacI-GFP signals with the distance ≤0.67 μm were regarded as paired, because we found that two juxtaposed homologous loci within this distance were already connected by a single RNA dot in smFISH (p2 state in Fig. 3). In smFISH experiments, the distance between two homologous loci with a single RNA dot was 0.53 μm in average (sd = 0.096, 32 cells), which corresponds to 0.67 μm in live cells after shrinkage correction.

**Chromatin immunoprecipitation sequencing (ChIP-seq) assay.** Diploid pat1-as2 strains expressing the GFP-fusion protein of Rhn1, Pab2, or Seb1 were used to induce synchronized meiosis[47] (Supplementary Table 2 for strain list). The cells were pre-cultured in YES (without adenine) then in EMM2-N for 3 h, and after addition of 1-NM-PP1 (Cayman Chemical Company, Ann Arbor, MI, USA), at a final concentration of 25 μM, they were cultured for 2 h. Approximately 50% of the cells were in meiotic prophase (which was checked by microscopic observation to count the elongated horsetail nucleus). The cells were then fixed with 1% formaldehyde for 30 min at 26 °C.

High-throughput sequencing was carried out using the HiSeq 2500 apparatus (Illumina, San Diego, CA, USA). Input and ChIP products were processed and sequenced according to the manufacturer's instructions. Briefly, DNA was sheared to an average size of ~250 bp by ultrasonication (Covaris, Woburn, MA, USA), end-repaired, ligated to sequencing adapters, amplified, size-selected, and sequenced, to generate single-end 65-bp reads. Full-length A.v. polyclonal GFP antibody (632460; Clontech, Mountain View, CA, USA) was used for ChIP analysis. The ChIP sequence data are available at the Sequence Read Archive, with accession number SRP129475. Fold enrichments (ChIP/WCE) of more than 1.5 are labeled in red on the map.

**Single molecular RNA fluorescence in situ hybridization (smFISH).** Vegetative growing cells were washed with EMM2-N medium. Cells (1 × 10⁶) in 0.3 mL EMM2-N were spread on an ME plate and incubated at 26 °C for ~8 h to induce meiosis. The cells (about 50–80% of the cells conjugated and were in meiotic prophase) were fixed with 4% formaldehyde (16% aqueous in 10 mL, in pre-scored ampules; PolySciences, Warrington, PA, USA) at room temperature (~25 °C) for 30 min. Single molecular RNA FISH was carried out using a protocol originally developed for Saccharomyces cerevisiae[48] and modified for Shizosaccharomyces pombe[49]. The RNA probe sets for sme2, omt3 (ncRNA130), and lncRNA584, listed in Supplementary Table 4, were custom-designed on the website of LGC Biosearch Technologies (Petaluma, CA, USA). The probes were co-synthetically labeled with

Quasar® 570 or Quasar® 670 (Stellaris®, LGC Biosearch Technologies). Chromatic shifts were corrected using the Chromagnon software[50].

**Reporting summary**. Further information on research design is available in the Nature Research Reporting Summary linked to this article.

## Data availability

The authors declare that the data supporting the findings of this study are available within the paper and its supplementary information files. The data of ChIP-seq analysis are available at the Sequence Read Archive, with accession number SRP129475. The Source Data underlying Figs. 1b, c, 3a–c, 4a, b, 5d and 6a and Supplementary Figs. 2e, 3, 4a–c, 5 and 7a are provided as a Source Data file. All data are available from the corresponding authors upon reasonable request.

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

## Acknowledgements

We are grateful to Y. Nagahama and C. Mori for technical assistance for experiments; T.G. Yamamoto and A. Matsuda for helpful discussions; Drs. N. Yokoda and R. Nakato for ChIP-seq data analysis; Drs. Y. Shichino, A. Yamashita, S. Hauf, D. Zenklusen for smFISH protocol; Dr. A. Hayashi for mutant characterization. This work was supported by KAKENHI grants from MEXT of Japan: JP26114725 and JP25291079 to D.-Q.D.;

JP18H05532 to J.-i.N.; JP15H05976, JP15H05970 and JP15K21761 to K.S.; JP25116006, JP18H05528, JP17H03636 to T.H.; JP17H01444 and JP18H05533 to Y.H.

## Author contributions

D.-Q.D., T.H., and Y.H. conceived and designed the experiments. D.-Q.D., K.O., Y.K., E.O., J.i.N., and Y.C. performed experiments. D.-Q.D., Y.K., K.S., J.i.N., Y.C., T.H., and Y.H. analyzed the data. All the authors contributed reagents/materials/analysis tools. D.-Q.D., T.H., and Y.H. wrote the manuscript with the input from all the authors.

## Competing interests

The authors declare no competing interests.
