## [Peer Review File · Nature Communications]

Reviewers' comments:

Reviewer #1 (Remarks to the Author):

The paper by Ding et al reports the identification of novel RNA-protein complexes that promote homolog pairing during *S. pombe* meiosis.

The strategy used was to screen for proteins colocalizing with the Sme2 RNA, an RNA previously shown to localize at its locus and to be important for pairing. 10 proteins were identified: they form foci with Sme2 RNA/locus and form also additional foci. By analyzing mutants, it was concluded that 6 of them are required for pairing at the Sme2 locus. In these mutants, the Sme2-RNA foci are affected, as they formed multiple dots instead of one as seen in wt. The authors then searched for the chromosomal binding sites of these proteins, as nine of them form foci at other locations than the Sme2 locus. For three of them (Rhn1, Pab2, and Seb1), ChIPSeq was performed. Using a powerful screening of a library of LacO insertions, the three Seb1 foci were mapped, one near the Sme2 locus, and two at additional loci. These two loci also show pairing at stage I of prophase. This pairing is Rec12 independent and depends on Rhn1 and Seb1. Comparing with Chip data, candidate genes were analyzed. Two non-coding RNA genes, Omt3 and ncRNA.584, were required for Seb1 focus formation. These two LncRNAs form often a single focus at their loci. The authors tested the effect of hexanediol on focus formation and found that it disrupted the foci (RNA, pairing, and proteins) in a reversible manner.

This is a very interesting study that follows the earlier pioneering work on the pairing at the Sme2 locus. This study generalizes the properties uncovered at the Sme2 locus, and importantly identifies proteins involved in this process. Interestingly the other loci also encode for LncRNAs and the foci are proposed to have phase separation properties based on hexanediol treatment.

The paper is clearly written, very concise and most of the conclusions are convincing. However, there are several important information missing which are necessary to build a fully consistent model. In addition, the presentation needs clarification for several analysis. In general, there is a lack of information and data about how the experiments were made or about data itself.

General comments, essential revisions.

1) It should be clarified from the beginning of the results, (and with the presentation of SupFig1), how many dots are detected for each protein, or at least for Rhn1, Pab2, and Seb1. One realizes later in the manuscript that they are 3 dots for Seb1. The data that establishes this quantification should be provided. It should also be explained how the dots are quantified (threshold values...). This is a very important point that is oversimplified in the text. In fact, if the interpretation of cytology and ChipSeq is correct, one may expect to have many more than 3 dots of Seb1, and the other weaker ones are also predicted to contribute to pairing. The authors should explain and comment on this.

2) The assay for pairing is described in the legend of fig1. This should be provided in the main text: it is essential for non-specialists to understand what is being looked at. More explanation is needed about the different substages, about the horsetail movement and how it relates to meiotic prophase and to other events during meiotic prophase, specifically meiotic recombination (ie when does recombination, initiation to resolution, occur with respect to pairing). It is specifically important from the beginning to explain the contribution of recombination such that one understands properly the pairing frequency across different substages.

3) L104. The colocalization of the Smp proteins with Seb1 should be quantified. It should also be clarified: do all Seb1 dots colocalize with Smps? Do all Smps dots colocalize with Seb1? At which stage? If the number of dots is clarified before this analysis (see comment above), it will avoid confusion.

4) Fig1b: Values of percent of pairing should have standard errors, and differences between mutants and wild type tested statistically. The legend indicates 20 to 30 cells were measured but how many duplicates? This comment applies to all pairing measurements. (see also comment 9).

5) The analysis of the effect of Smp mutant on Sme2 RNA dots should be clarified. At which stage is this analyzed? When 2 dots are seen, are these on the two homologs? Do Sme2 RNA still localize at the Sme2 locus?

6) Additional information is required with respect to the role of the Smp proteins for pairing: Are they mutually dependent for their localization? At least some of the most important ones should be tested: What is the effect of Seb1, Rhn1 or Pab2 mutants on other Smp dots? Importantly, if the hypothesis proposed is correct, one expects that these mutants would have a strong global (not only locus-specific) phenotype on pairing. The authors need to test this, which also requires to introduce the Rec12 mutation to separate the contribution of recombination. As the identification of these proteins is one of the major advances with respect to previous analysis on Sme2, this is an essential point. Also one needs to know what is their meiotic phenotype (divisions, viability).

7) ChIP seq. This analysis is very poorly documented. This must be fully revised: it is not sufficient to mention that 50 binding sites were identified and only to provide a table with the list. The authors have to show the data that drives them to this conclusion. This implies providing data for vegetative cells and meiotic cells, showing negative control, explaining how peaks were called, how intensities were computed? How were the 50 peaks selected, among a total of how many? Among the peaks detected, one needs to know and understand if they are meiotic-specific and what is the overlap between the three data sets. Some data is presented in Table 1, but this is largely insufficient. Some peaks are at centromeres, this should be discussed. How many duplicates?

Actually, presenting the ChIP data before the LacO screen does not help. I would suggest moving the ChIP data presentation to the beginning of the next section, L142.

8) Fig2d shows that a fraction (60% at stage I) of nuclei is not paired at the A55 and C24 loci. What is the localization of Smps (Seb1, Rhn1, Pab2) in such context? Does it form one focus on each homolog? Also, the criteria for pairing is different than on Fig 1 (as explained later in the text). This should be explained when the data of Fig2 is presented. It is quite clear from Fig2b that the loci are next to each other, but not overlapping.

9) A general comment about pairing: As the authors measure distances and have the data, they should provide graphs with distance distribution so that one understands what is the magnitude of the effect. The information about percent of pairing is not sufficient. This is particularly important, because, as explained in the methods, they had to change the threshold for A55, because of this doublet configuration.

10) Fig2e: The ChipSeq peaks near A55 and C24 should be highlighted in Suptable 1. Actually, the ncRNA.584 is not in the list of the Pab2 ChipSeq, whereas Omt3 and Sme2 are the strongest peaks. What is the explanation?

11) Fig2f and G. The result of the deletion of the Omt3 gene on Seb1 occupancy is trivial in a way because the sequences where the protein binds have been deleted (quantification needed: Seb1 focus analyzed in how many nuclei?). However, A55 still forms a closed doublet? Is this observed in all nuclei (quantification needed)? How is this possible? (same question for Ski3 deletion, panel g). Maybe there are other weaker binding sites next by? If this is the interpretation of the authors? Then this unexpected result highlights again the need to integrate fully the other potential sites detected by ChipSeq.

Please move the yellow arrow slightly away, as it partially masks the dots.

In principle, it would have been more important to test the effect of Omt3 and ncRNA.584

deletion on pairing when heterozygous (but if the loci are paired in the homozygous deletion as this reviewer understands, it does not apply). As observed at *Sme2*, heterozygosity should abolish pairing.

12) In Fig3d, the *seb1* mutant seems to have a stronger effect on *Omt3* pairing than on *Sme2*? What is the interpretation? I guess it is possible that the protein complexes are not exactly identical at all loci. This is also why the question raised in point 6 is important.

13) Have the authors tested and identified any other LcnRNA expressing locus that does not pair efficiently at stage I?

14) The proposition of a phase separation property is interesting but needs more documentation: importantly, the authors should test that axis is not affected by hexanediol treatment. Of particular relevance is to show that the cohesin axis proteins (*Rec8* and *Pds5*), previously shown by the authors to be important, are not disrupted. The authors could discuss current literature showing phase separation properties at P granules which include many RNA processing proteins, including some identified here (the literature is very large; see for instance Sachdev et al. *Elife*. 2019 Jan 16;8). The pairing foci could thus also be P granules.

15) The movies are not helpful without additional guidance and labeling to know what we should be looking at and to identify the frames corresponding to hexanediol addition and removal. Some legends or maybe still images should help.

Specific comments

Table 1: the name of the *S. pombe* genes should be highlighted for easier identification

Methods: The degron strategy is described but it should be better documented: the evidence for depletion should be shown.

Some more information and explanation about the *ts Seb1* mutant is needed: What was the temperature used?

SupFig1. It will help to have the names of the genes indicated above the panel "a" instead of the numbers 1 to 10.

L105: The remaining Sm proteins, change to: The remaining three Sm proteins

L111. At this level of the analysis, the author should only conclude on a correlation between pairing phenotype and *Sme2* RNA foci.

Fig2b: On this image as well on other images, please indicate the prophase stage.

Fig2d: *Rhn1* mutant has no effect at the C24 locus at stage I. What is the interpretation?

L185: Can you develop what you mean about the analogy between pairing and liquid droplets?

Fig3c: SPNCRNA and LacI GFP foci do not look overlapping by eye, although the merge is OK. This is puzzling.

Reviewer #2 (Remarks to the Author):

This manuscript describes identification of RNA-protein complexes that promote meiotic homologous chromosome pairing at specific loci. Previous studies including the ones from the authors' group showed that RNA-protein complex composed of lncRNA *sme2* and RNA-binding protein *Mei2* localizes at the *sme2* locus on chromosome II and promotes homologous chromosome pairing there. In this manuscript, the authors now provide a comprehensive list of proteins and lncRNAs that promote homologous pairing on chromosomes including other than chromosome II. First, they identified a set of proteins that exhibit dot-like structures in the meiotic prophase nucleus through visual screening. Six of these proteins were required for robust pairing of the *sme2* locus. By chIP-seq of these proteins, they determined chromosome loci that these proteins localize to. This approach identified A55 (on chromosome I) and C24 (on chromosome III), in addition to *sme2* (on chromosome II). They found that these loci produce lncRNAs, and the

deletion of these sites delays homologous pairing around these loci. Finally, they show that the RNA-protein 'dots' in the meiotic nucleus exhibit phase-separation-like behaviour upon 1,6-hexanediol treatment.

Overall, this study nicely extends their previous finding that RNA-protein complex promotes homologous pairing at specific loci on chromosome II, and now provides a generalized view that each of chromosomes carries at least one major site for RNA-protein-mediated efficient homologous pairing. The data are beautifully presented and the conclusion is clear. I feel this study gives a significant advance to our understanding of the mechanisms for homologous chromosome pairing. I would support publication in Nature Communications if the authors appropriately address the points raised below.

Major points:

- Do the newly identified loci (A55 and C24) show earlier pairing compared to other loci on same chromosomes? If not, these sites may not be necessarily 'a primary "pairing center"' (page 11, line 234).

Minor points:

- Requirement of Seb1 seems to be different among chromosomes (Figure 3d). Any explanations?

Reviewer #3 (Remarks to the Author):

Ding et al. report that lncRNA-protein complexes, which form at specific loci expressing these lncRNAs, mediate recombination-independent pairing of homologous chromosomes during meiosis in *S. pombe*. In a series of elegantly designed genetic and genomic experiments the authors identified conserved Smp (sme2 RNA-associated protein) factors that are required for efficient pairing at the sme2 locus on chromosome II and two new loci (A55 and C24) mediating chromosome I and III pairing, respectively. They go on to show that A55 and C24 encode omt3 and SPNCRNA.584 lncRNAs that are important for homologous chromosome pairing and that accumulate in single RNA dots at their respective loci. Finally, the authors analysed phase separation properties of the lncRNA-protein complexes using 1,6-hexanediol, which interferes with hydrophobic RNA-protein interactions in membrane-less granules. They found that the compound abolished Smp protein foci in a reversible manner.

Based on their findings the authors propose a persuasive model positing that homologous chromosomes undergo a sequential alignment process during meiosis. This process involves initial chromosome pairing mediated by locus-specific lncRNA-protein complexes, followed by stable chromosome association via homologous recombination. Given that Smp proteins are conserved and numerous lncRNAs are specifically induced during gametogenesis across species this model likely pertains to meiosis in higher eukaryotes.

A critical role for developmental-stage specific RNA/protein complexes in controlling chromosome behaviour raises the important question why sme2, omt3 and SPNCRNA.584 are meiosis-specific. The Mmi1-dependent mechanism of meiotic mRNA degradation during mitotic growth is well studied but are meiotic lncRNAs controlled in the same way? Mmi1 was reported to show a moderate affinity for sme2's meiRNA, which argues in favor of a model based on RNA stability rather than promoter-mediated transcriptional regulation (Harigaya et al., Nature 2006). It would be interesting to learn more about the developmental regulation of the new pairing lncRNAs identified in this study. How are omt3 and SPNCRNA.584 controlled in mitosis and meiosis? Do they contain determinant of selective removal (DRS) motifs? In this context it is noteworthy that the catalytic exosome subunit Rrp6 appears to target SPNCRNA.584 but not omt3, which indicates that they are not regulated in the same way (Mukherjee et al., RNA 2016; supplemental table Rrp6 targets). This could be related to the fact that SPNCRNA.584 is an antisense lncRNA.

Minor comments:

The arrow-heads in Fig. 2f and g appear to partially cover the dots they are meant to highlight and should therefore be moved up a bit.

Should the authors not rename the newly discovered pairing-mediator SPNCRNA.584 into something more appealing (such as pal584 for pairing lncRNA 584)?

We appreciate the constructive comments of the editor and reviewers. Following the comments, we have carried out the suggested experiments or clarified data presentation. New data of suggested experiments have been included in Figure 6 and Supplementary Figures 1, 2, 3, 4, 5, 6d, and 7. Point-by-point responses are shown below. *Blue italic sentences* are reviewer's comments, black roman sentences are our responses, and *red font parts* are quotation from the text in this response letter. The response letter is followed by the manuscript with the revised parts marked in red font.

Reviewer #1: General comments, essential revisions.

1) It should be clarified from the beginning of the results, (and with the presentation of SupFig1), how many dots are detected for each protein, or at least for Rhn1, Pab2, and Seb1. One realizes later in the manuscript that they are 3 dots for Seb1. The data that establishes this quantification should be provided. It should also be explained how the dots are quantified (threshold values...). This is a very important point that is oversimplified in the text. In fact, if the interpretation of cytology and ChipSeq is correct, one may expect to have many more than 3 dots of Seb1, and the other weaker ones are also predicted to contribute to pairing. The authors should explain and comment on this.

(Response 1) Following this comment, we have added data on the number of Smp dots in Supplementary Fig. 1b (lines 91-92), and have described the method for counting the number of Smp dots: "The "dots" were identified using the 2D polygon finder function of the softWoRx software (GE Healthcare); the fluorescence intensity threshold of a "dot" was set as more than 1.5 times higher than the average intensity in the nucleus." in the legend to Supplementary Fig. 1b. The weaker ones were not obvious because of the high background nuclear staining and were difficult to quantify, so they are excluded from counting.

2) The assay for pairing is described in the legend of fig1. This should be provided in the main text: it is essential for non-specialists to understand what is being looked at. More explanation is needed about the different substages, about the horsetail movement and how it relates to meiotic prophase and to other events during meiotic prophase, specifically meiotic recombination (ie when does recombination, initiation to resolution, occur with respect to pairing). It is specifically important from the beginning to explain the contribution of recombination such that one understands properly the pairing frequency across different substages.

(Response 2) We appreciate this comment to improve readability. In *S. pombe*, the horsetail period mostly corresponds to the meiotic prophase. However, typical meiotic stages such as leptotene or zygotene are not cytologically obvious and no synaptonemal complex forms in *S. pombe*. We have provided this information in the Introduction (lines 64-

68). In order to follow the progression through meiotic prophase in these circumstances, we divided the horsetail period into 5 substages by time in each of living cells. We explained how the substages were defined in lines 105-108.

3) L104. The colocalization of the Smp proteins with Seb1 should be quantified. It should also be clarified: do all Seb1 dots colocalize with Smps? Do all Smps dots colocalize with Seb1? At which stage? If the number of dots is clarified before this analysis (see comment above), it will avoid confusion.

(Response 3) Following the comment, we quantified the Smp dots colocalized with Seb1 in the horsetail stage as shown in Supplementary Fig.1c (lines 113-114).

4) Fig1b: Values of percent of pairing should have standard errors, and differences between mutants and wild type tested statistically. The legend indicates 20 to 30 cells were measured but how many duplicates? This comment applies to all pairing measurements. (see also comment 9).

(Response 4) The distance measurements were repeated to the number of cells indicated (20 – 30 in each strain). To provide the statistics of distance distribution in 20 to 30 cells, we have presented the data in the form of a percentile rank plot of the distance between homologous loci (Fig. 6a, Supplementary Fig. 3, 5, 7a). The percentile rank plot directly represents the statistics of distance distribution, and has been generally accepted in previous publications in our and other groups (e.g., Weiner & Kleckner, Cell 1994).

5) The analysis of the effect of Smp mutant on Sme2 RNA dots should be clarified. At which stage is this analyzed? When 2 dots are seen, are these on the two homologs? Do Sme2 RNA still localize at the Sme2 locus?

(Response 5) We thank you for this comment. The analysis was carried out in an asynchronous population of horsetail nuclei. To clarify the stage, we have added the phrase “the horsetail stage” in the legend to Fig. 1c. When 2 dots are seen, these are likely localized at the homologous *sme2* loci. With this regard, we have added the following statements in the legend to Fig. 1c: **“Frequency (%) of the number of *sme2* RNA dots in WT and various Smp-defective mutant cells at the horsetail stage. The RNA dots were classified into 4 classes (“1” for a single dot, “2E” for 2 RNA dots with equal size, “2U” for 2 RNA dots with unequal size, and “>3” for more than 2 dots). Representative images for one to three or more dots are shown on the left. Two dots with equal size (2E) or unequal size (2U) were counted separately. Class “2E” likely represents *sme2* RNA accumulated at each of the homologous *sme2* loci. More than 50 horsetail nuclei in an asynchronous population were observed for each strain, in three independent experiments.”**

6) Additional information is required with respect to the role of the Smp proteins for pairing:
Are they mutually dependent for their localization? At least some of the most important ones should be tested: What is the effect of *Seb1*, *Rhn1* or *Pab2* mutants on other Smp dots?

(Response 6a)

We agree that this is an important question. We speculate that Smp proteins form heterogeneous complexes, and roles for Smp proteins may vary from one site to another, depending on each RNA involved. To address this question, extensive biochemical analysis will be required. This issue is important to address, but beyond the scope of this paper. To avoid confusion that Smp proteins form a homogeneous complex, we have deleted “possibly forming a complex” (lines 113-114).

*Importantly, if the hypothesis proposed is correct, one expects that these mutants would have a strong global (not only locus-specific) phenotype on pairing. The authors need to test this, which also requires to introduce the *Rec12* mutation to separate the contribution of recombination. As the identification of these proteins is one of the major advances with respect to previous analysis on *Sme2*, this is an essential point. Also one needs to know what is their meiotic phenotype (divisions, viability).*

(Response 6b)

We appreciate this comment. We have added new data in Fig. 4, showing the contributions of recombination to lncRNA-mediated pairing. Also to avoid confusion, we avoided using the term “a primary pairing center”, and discussed possible models in the Discussion (lines 312-324): “Our findings demonstrate that ----- that juxtapose a pair of homologous chromosomes^{31,32}.”

7) ChIP seq. This analysis is very poorly documented. This must be fully revised: it is not sufficient to mention that 50 binding sites were identified and only to provide a table with the list. The authors have to show the data that drives them to this conclusion. This implies providing data for vegetative cells and meiotic cells, showing negative control, explaining how peaks were called, how intensities were computed? How were the 50 peaks selected, among a total of how many? Among the peaks detected, one needs to know and understand if they are meiotic-specific and what is the overlap between the three data sets. Some data is presented in Table 1, but this is largely insufficient. Some peaks are at centromeres, this should be discussed. How many duplicates? Actually, presenting the ChIP data before the *LacO* screen does not help. I would suggest moving the ChIP data presentation to the beginning of the next section, L142.

(Response 7) We greatly appreciate this comment. As suggested, we have now presented the ChIP-seq data after the *lacO* data (lines 143-146). We have provided the entire data set

of ChIP-seq along the genome as Supplementary Material to replace Supplementary Table 1 in the original version. The ChIP-seq experiments were carried out for three Smp proteins, and one time for each protein. Although binding peaks at the centromere are present in a long range spanning the centromere region, the peak height is lower than the three IncRNAs regions as can be seen in the Supplementary Material.

8) Fig2d shows that a fraction (60% at stage I) of nuclei is not paired at the A55 and C24 loci. What is the localization of Smps (Seb1, Rhn1, Pab2) in such context? Does it form one focus on each homolog? Also, the criteria for pairing is different that on Fig 1 (as explained later in the text). This should be explained when the data of Fig2 is presented. It is quite clear from Fig2b that the loci are next to each other, but not overlapping.

(Response 8) Criteria for pairing were defined by smFISH. Therefore, we moved the data of pairing (now in Fig. 4) to after smFISH (Fig. 3), and described the criteria in the text when presenting the data of pairing in Fig. 4 (lines 183-188 and 200-203 in Results; also lines 379-385 in Methods):

(lines 183-188) “In the smFISH experiments, the distance between two homologous loci connected by a single RNA dot was 0.53 μm on average (sd=0.096, 32 cells). Because the cells shrank to 79% by fixation compared to live cell, this distance corresponds to 0.67 μm in live cells. Therefore, based on the pairing distance of RNA dots, the LacI-GFP signals with a distance of $\leq 0.67 \mu\text{m}$ (“p1” and “p2” states) were hereafter regarded as “paired”.”

(lines 200-203) “We measured the pairing frequency at the newly identified loci (A55 and C24) in the horsetail substages I-V to compare with that at the *sme2* locus (Fig. 4). Pairing frequency was determined with a distance of $\leq 0.67 \mu\text{m}$ as paired based on the pairing distance of RNA dots.”

(lines 379-385) “For counting the pairing frequency in Fig. 2d and Fig. 4d, LacI-GFP signals with the distance $\leq 0.67 \mu\text{m}$ were regarded as “paired”, because we found that two juxtaposed homologous loci within this distance were already connected by a single RNA dot in smFISH (“p2” state in Fig. 3). In smFISH experiments, the distance between two homologous loci with a single RNA dot was 0.53 μm in average (sd=0.096, 32 cells), which corresponds to 0.67 μm in live cells after shrinkage correction.”

Thus, we believe that the criteria are now clear.

9) A general comment about pairing: As the authors measure distances and have the data, they should provide graphs with distance distribution so that one understands what is the magnitude of the effect. The information about percent of pairing is not sufficient. This is particularly important, because, as explained in the methods, they had to change the threshold for A55, because of this doublet configuration.

(Response 9) Thank you for the comment. Following this comment, we presented distance distribution as a percentile rank plot of the distance between homologous loci as described in Response 4 above.

10) Fig2e: The ChipSeq peaks near A55 and C24 should be highlighted in Suptable 1. Actually, the ncRNA.584 is not in the list of the Pab2 ChipSeq, whereas Omt3 and Sme2 are the strongest peaks. What is the explanation?

(Response 10) Selection of CHIP-seq peaks depends on thresholding. Thus, we have deleted Supplementary Table 1, and instead have presented the entire map of CHIP-seq results in Supplementary material.

11) Fig2f and G. The result of the deletion of the Omt3 gene on Seb1 occupancy is trivial in a way because the sequences where the protein binds have been deleted (quantification needed: Seb1 focus analyzed in how many nuclei?). However, A55 still forms a closed doublet? Is this observed in all nuclei (quantification needed)? How is this possible? (same question for Ski3 deletion, panel g). Maybe there are other weaker binding sites next by? If this is the interpretation of the authors? Then this unexpected result highlights again the need to integrate fully the other potential sites detected by ChipSeq.

(Response 11a) This result is important because it provides direct evidence to indicate the DNA sequences. i.e., the *omt3* gene, responsible for the accumulation of Smp proteins. Fig. 2f and g are included to show whether Smp accumulates or not, and it does not matter whether they are paired or unpaired. We have described possible contributions of weaker binding sites as revealed by CHIP-seq to pairing in the Discussion (lines 312-324): “Our findings demonstrate that ----- that juxtapose a pair of homologous chromosomes^{31,32}.”

Please move the yellow arrow slightly away, as it partially masks the dots.

(Response 11b) As suggested, we have moved the arrows. Thank you.

In principle, it would have been more important to test the effect of Omt3 and ncRNA.584 deletion on pairing when heterozygous (but if the loci are paired in the homozygous deletion as this reviewer understands, it does not apply). As observed at Sme2, heterozygosity should abolish pairing.

(Response 11c) Thank you for this important comment. We have added new experiments involving heterozygosity. In these experiments, we observed pairing in the heterozygous strain bearing the *sme2* and *omt3* genes at the *omt3* locus. The results demonstrated that no robust pairing occurred in this heterozygous strain (Fig. 6a) and no fusion was observed between *sme2* RNA and *omt3* RNA foci as shown in Fig. 6b (lines 259-280).

12) In Fig3d, the seb1 mutant seems to have a stronger effect on Omt3 pairing than on Sme2? What is the interpretation? I guess it is possible that the protein complexes are not exactly identical at all loci. This is also why the question raised in point 6 is important.

(Response 12) Thank you for the comment. We agree that the protein complexes are not exactly identical at all loci, but instead heterogeneous depending on RNA involved. In fact, sme2, A55, and C24 seem to have different properties in pairing. sme2 RNA is the most robust, omt3 RNA is intermediate, and lncRNA584 is weak in the promotion of pairing as shown in Fig. 4, Fig. 5d, and Fig. 6a; omt3 RNA and lncRNA584 depend on Smp proteins more strongly than the sme2 RNA for their chromosomal accumulation (Fig. 3d). As we have no clear evidence to explain these differences at present, we did not make any statements in this paper. Also see Response 6a above.

13) Have the authors tested and identified any other LcnRNA expressing locus that does not pair efficiently at stage I?

(Response 13) We feel that this point is beyond the scope of the present paper. However, we will examine whether other lncRNAs affect pairing in our future work.

14) The proposition of a phase separation property is interesting but needs more documentation: importantly, the authors should test that axis is not affected by hexanediol treatment. Of particular relevance is to show that the cohesin axis proteins (Rec8 and Pds5), previously shown by the authors to be important, are not disrupted.

(Response 14a) This is an important point. We have added experiments on the effect of hexanediol treatment on Rec8. Rec8 is retained on chromatin in the presence of 1,6-hexanediol (Supplementary Fig. 6d; lines 253-255).

The authors could discuss current literature showing phase separation properties at P granules which include many RNA processing proteins, including some identified here (the literature is very large; see for instance Sachdev et al. Elife. 2019 Jan 16;8). The pairing foci could thus also be P granules.

(Response 14b) Thank you for the suggestion. We have added statements about RNA bodies including P granules in the Discussion (lines 329-333).

15) The movies are not helpful without additional guidance and labeling to know what we should be looking at and to identify the frames corresponding to hexanediol addition and removal. Some legends or maybe still images should help.

(Response 15) As suggested, we have improved the visualization by adding screen scripts on the movies.

Specific comments

Table 1: the name of the S. pombe genes should be highlighted for easier identification

(Response 16) As suggested, we have moved the gene names to the first column, and made them bold.

Methods: The degron strategy is described but it should be better documented: the evidence for depletion should be shown.

(Response 17) As suggested, we have presented the evidence for degron depletion in Supplementary Fig. 2a,b.

Some more information and explanation about the ts Seb1 mutant is needed: What was the temperature used?

(Response 18) We have described characterization of the *seb1* mutant in Supplementary Fig. 2c-f and its legend. We also renamed the *seb1* mutant allele to *seb1-E38*. We observed all strains including this strain at 26°C as described in Methods (lines 374-375).

SupFig1. It will help to have the names of the genes indicated above the panel "a" instead of the numbers 1 to 10.

(Response 19) As suggested, we have added the gene names in the figure. We also moved this panel (Supplementary Fig. 1a) to main Fig. 1a.

L105: The remaining Sm proteins, change to: The remaining three Sm proteins

(Response 20) We have revised as suggested (line 114).

L111. At this level of the analysis, the author should only conclude on a correlation between pairing phenotype and Sme2 RNA foci.

(Response 21) As suggested, we have rephrased the text to "**These results indicate that Smp proteins mediate chromosomal accumulation of *sme2* RNA dots, which correlates with robust pairing.**" (lines 122-124).

Fig2b: On this image as well on other images, please indicate the prophase stage.

(Response 22) We defined the meiotic prophase stage in *S. pombe* in the Introduction as described in Response 2. In live-cell observation for pairing, we subdivided the horsetail period into 5 substages in the time-course. In other observations, "the horsetail" represents an asynchronous population of the meiotic prophase. We described the stages in the legend to each figure.

Fig2d: Rhn1 mutant has no effect at the C24 locus at stage I. What is the interpretation?

(Response 23) The effect of Smp on C24 pairing probably reflects the presence of the nucleolus. We discussed the effect of the nucleolus in the Results (lines 215-224): “In contrast, IncRNA584 had limited contributions to pairing at C24 in both recombination positive or negative background (“ Δ IncRNA584” in Fig. 4a and Supplementary Fig. 5). The depletion or mutation of Rhn1 and Seb1 decreased the pairing frequency at the C24 locus (Fig. 4b), indicating that C24 pairing requires Smp proteins. Therefore, it is likely that Smp-dependent pairing at nearby regions contributes to pairing at the C24 locus. The C24 locus is close to the rDNA repeat cluster located at both ends of Chromosome III, which form the nucleolus. Pairing of chromosome III is likely dominated by the gigantic RNA body of the nucleolus, underrating the contributions of IncRNA584 to the pairing of homologous chromosomes.”

L185: Can you develop what you mean about the analogy between pairing and liquid droplets?

(Response 24) We have added statements about the analogy between pairing and liquid droplets in lines 233-235: “In smFISH staining of these three IncRNA species, they always appear as a round shape with different sizes (“p2” and “p1” in Figure 3a-c). Furthermore, no two RNA dots touching each other were observed, implying an immediate fusion to one dot.”

Fig3c: SPNCRNA and Lacl GFP foci do not look overlapping by eye, although the merge is OK. This is puzzling.

(Response 25) SPNCRNA and Lacl GFP foci are closely associated, but not exactly overlapping with each other. So this appearance is real.

Reviewer #2:

Major points:

- Do the newly identified loci (A55 and C24) show earlier pairing compared to other loci on same chromosomes? If not, these sites may not be necessarily ‘a primary “pairing center”’ (page 11, line 234).

(Response 26) We greatly appreciate this important comment. A55 and C24 do show earlier pairing like *sme2*: we measured the pairing frequency at about 150 sites along the whole genome in the presence or absence of recombination. The effect of these loci on pairing is limited to a local region; even the strongest *sme2* locus promotes pairing only within approximately 200-kb regions as reported previously (Ding et al., 2012). Thus, we avoided the use of the term “a primary pairing center”, and discussed possible models in

the Discussion (lines 312-324): “Our findings demonstrate that ----- that juxtapose a pair of homologous chromosomes^{31,32}.”

Minor points:

- Requirement of Seb1 seems to be different among chromosomes (Figure 3d). Any explanations?

(Response 27) Thank you for the comment. We speculate that Smp complexes are not exactly identical at all loci, but instead heterogeneous depending on RNA involved. In fact, *sme2*, *A55*, and *C24* seem to have different properties in pairing. *sme2* RNA is the most robust, *omt3* RNA is intermediate, and *lncRNA584* is weak in the promotion of pairing as shown in Fig. 4, Fig. 5d, and Fig. 6a; *omt3* RNA and *lncRNA584* depend on Smp proteins more strongly than the *sme2* RNA for their chromosomal accumulation (Fig. 3d). As we have no clear evidence to explain these differences at present, we did not make any statements in this paper.

Reviewer #3 (Remarks to the Author):

*A critical role for developmental-stage specific RNA/protein complexes in controlling chromosome behaviour raises the important question why *sme2*, *omt3* and *SPNCRNA.584* are meiosis-specific. The *Mmi1*-dependent mechanism of meiotic mRNA degradation during mitotic growth is well studied but are meiotic lncRNAs controlled in the same way? *Mmi1* was reported to show a moderate affinity for *sme2*'s *meiRNA*, which argues in favor of a model based on RNA stability rather than promoter-mediated transcriptional regulation (Harigaya et al., Nature 2006). It would be interesting to learn more about the developmental regulation of the new pairing lncRNAs identified in this study. How are *omt3* and *SPNCRNA.584* controlled in mitosis and meiosis? Do they contain determinant of selective removal (DRS) motifs? In this context it is noteworthy that the catalytic exosome subunit *Rrp6* appears to target *SPNCRNA.584* but not *omt3*, which indicates that they are not regulated in the same way (Mukherjee et al., RNA 2016; supplemental table *Rrp6* targets). This could be related to the fact that *SPNCRNA.584* is an antisense lncRNA.*

(Response 28) We greatly appreciate this important comment. To address this question, we carried out experiments to measure the effect of the DSR motif in pairing and found that this motif did not affect pairing frequency. We have added this result in Supplementary Fig. 7 and lines 300-311 in the text with the citation of Mukherjee et al., RNA 2016: “Notably, the identified Smp proteins are all meiosis-specific transcripts. --- This is consistent with our finding that DSR is not involved in lncRNA-mediated homologous pairing.”

Minor comments:

The arrow-heads in Fig. 2f and g appear to partially cover the dots they are meant to highlight and should therefore be moved up a bit.

(Response 29) Thank you for pointing this out. We have moved the arrows in the revised manuscript.

Should the authors not rename the newly discovered pairing-mediator SPNCRNA.584 into something more appealing (such as pal584 for pairing lncRNA 584)?

(Response 30) Thank you for the suggestion. Accordingly, we have re-named this RNA to “lncRNA584” throughout the manuscript.

Reviewers' comments:

Reviewer #1 (Remarks to the Author):

The authors have answered most questions and performed additional experiments. 3 points remain to be clarified.

1) In pairing distances data reported in supFig3, 5, 7 and fig6, please label on the Y axis the value corresponding to pairing as a single dot (0.53um?).

Could you clarify the number of points plotted: It is mentioned that more than 20 cells were counted. If all stages are included, this would mean in average 4 cells per stage? Is it correct? It is also mentioned that the total point number of distances in each stage were more than 100. How is this possible? Some normalization has probably been performed to homogenize datasets with different cell numbers and thus to plot the percentile rank. It should be explained how, but providing a number of points that do not correspond to the data is misleading.

2) It is surprising that ChIP data does not have duplicates. As for any other data, reproducibility is an essential requirement of experimental sciences. I do not think this is compatible with publication in a high standard journal.

3) In sup fig1 panel b, it would help to label the four rows with the corresponding proteins. The legend indicates graphs for Seb1-GFP, Rhn1-GFP, and Pab2-GFP, but only two graphs are shown?

Reviewer #2 (Remarks to the Author):

My concerns have been appropriately addressed. I support publication.

Reviewer #3 (Remarks to the Author):

Ding et al. have substantially revised their manuscript by including new data. They now report that omt3 and lncRNA584 also contain DSR motifs and show that 14 DSR repeats do not affect the homologous pairing frequency when inserted at the A37 locus. I presume the presence of DSR motifs in omt3 and lncRNA584 means that these lncRNAs are down-regulated in mitosis via Mmi1, which answers my question.

Minor comments:

Page 14, line 300: ...Smp proteins are all encoded by meiosis-specific transcripts.

P14, line 303: ...contained in such RNAs, including sme2.

P15, line 309: please replace "SPNCRNA.584" by "lncRNA584".

The authors write: "The exosome targets only SPNCRNA.584, but not omt3 RNA. This is consistent with our finding that DSR is not involved in lncRNA mediated homologous pairing". I am not sure I understand the logic of the last phrase.

Point-by-point responses to comments are described below. Blue italic fonts indicate referees' comments, followed by our responses in black roman fonts. Red fonts indicate quotation from the text. The response letter is followed by the manuscript with the revised parts marked in red font.

Reviewer #1 (Remarks to the Author):

The authors have answered most questions and performed additional experiments.

3 points remain to be clarified.

1) In pairing distances data reported in supFig3, 5, 7 and fig6, please label on the Y axis the value corresponding to pairing as a single dot (0.53um?).

Thank you for this comment. Following this comment, we drew a dotted line at 0.67 μm on the Y axis in supFig. 5, 7 and Fig. 6a, and labeled "0.35 μm " on the Y axis in supFig. 3. We also added the following statements in the legends to supFig. 5, 7 and Fig. 6a: "The dotted line at 0.67 μm on the Y axis indicates the threshold of pairing"; legends to supFig. 3: "The threshold of pairing is 0.35 μm as labeled on the Y axis."

Could you clarify the number of points plotted: It is mentioned that more than 20 cells were counted. If all stages are included, this would mean in average 4 cells per stage? Is it correct? It is also mentioned that the total point number of distances in each stage were more than 100. How is this possible? Some normalization has probably been performed to homogenize datasets with different cell numbers and thus to plot the percentile rank. It should be explained how, but providing a number of points that do not correspond to the data is misleading.

Thank you for pointing out our ambiguous statement about the number of measurements. We observed the same living cells continuously throughout the 5 substages, and the distance was measured at typically 5 timepoints. Thus, the total number of measurements in each dataset is the number of cells multiplied by the number of timepoints in each substage. Datasets with different cell numbers were normalized as a percentage. To avoid confusion, we added the following statement in Methods (lines 377-384):

"In measuring homologous chromosome pairing, we observed the same living cells continuously, and images were recorded every 5 minutes for 2-3 hours during the horsetail stage. We divided the horsetail stage equally to 5 substages. The number of cells observed was 20 – 40 for each strain. Distances were measured in a data set of time-lapse images composed of typically 5 timepoints in 20 – 40 cells, making about 100 – 200 measurements in each substage for each strain. The number of measurements was normalized as a percentage, and a percentile rank of the distance was plotted as described previously¹⁰."

2) It is surprising that ChIP data does not have duplicates. As for any other data, reproducibility is an essential requirement of experimental sciences. I do not think this is compatible with publication in a high standard journal.

The purpose of the ChIP-seq assay is to narrow the genomic sequences responsible for the robust pairing. The ChIP-seq assay simply provides screening of candidates for the further analysis. These screened candidate sequences were confirmed by the deletion analysis. ChIP-seq experiments were repeated with three co-localizing proteins in meiosis and vegetative growth. Reproducible results were obtained for three proteins. We believe that this is sufficient for a screening purpose. Presenting the whole-genome ChIP-seq results as supplements are beneficial for readers who are interested in those results.

3) In sup fig1 panel b, it would help to label the four rows with the corresponding proteins. The legend indicates graphs for Seb1-GFP, Rhn1-GFP, and Pab2-GFP, but only two graphs are shown?

Thank you for pointing out our error in the legend. We labeled sup Fig.1b with “Seb1-GFP” and “Rhn1-GFP”, and deleted “Pab2-GFP” in the legend: “Examples of projection images for **Seb1-GFP and Rhn1-GFP** (left). The number distribution of **Seb1-GFP and Rhn1-GFP** dots (right).”

Reviewer #2 (Remarks to the Author):

My concerns have been appropriately addressed. I support publication.

Thank you.

Reviewer #3 (Remarks to the Author):

Ding et al. have substantially revised their manuscript by including new data. They now report that omt3 and IncRNA584 also contain DSR motifs and show that 14 DSR repeats do not affect the homologous pairing frequency when inserted at the A37 locus. I presume the presence of DSR motifs in omt3 and IncRNA584 means that these lncRNAs are down-regulated in mitosis via Mmi1, which answers my question.

Minor comments:

Page 14, line 300: ...Smp proteins are all encoded by meiosis-specific transcripts.

P14, line 303: ...contained in such RNAs, including sme2.

P15, line 309: please replace “SPNCRNA.584” by “lncRNA584”.

Thank you. We changed as suggested.

The authors write: "The exosome targets only SPNCRNA.584, but not omt3 RNA. This is consistent with our finding that DSR is not involved in lncRNA mediated homologous pairing". I am not sure I understand the logic of the last phrase.

We rephrased to "It is also unlikely that the exosome is involved in lncRNA-mediated homologous pairing because *omt3* RNA is not listed as targets of the exosome²⁹" in lines 309-311.

REVIEWERS' COMMENTS:

Reviewer #1 (Remarks to the Author):

The authors have answered to all questions raised